# Myokines May Be the Answer to the Beneficial Immunomodulation of Tailored Exercise—A Narrative Review

**DOI:** 10.3390/biom14101205

**Published:** 2024-09-25

**Authors:** Zheng Lu, Zhuo Wang, Xin-An Zhang, Ke Ning

**Affiliations:** College of Exercise and Health, Shenyang Sport University, Shenyang 110102, China; luzheng20113@163.com (Z.L.); wangzhuo@syty.edu.cn (Z.W.)

**Keywords:** exercise, myokines, immune cells, skeletal muscle

## Abstract

Exercise can regulate the immune function, activate the activity of immune cells, and promote the health of the organism, but the mechanism is not clear. Skeletal muscle is a secretory organ that secretes bioactive substances known as myokines. Exercise promotes skeletal muscle contraction and the expression of myokines including irisin, IL-6, BDNF, etc. Here, we review nine myokines that are regulated by exercise. These myokines have been shown to be associated with immune responses and to regulate the proliferation, differentiation, and maturation of immune cells and enhance their function, thereby serving to improve the health of the organism. The aim of this article is to review the effects of myokines on intrinsic and adaptive immunity and the important role that exercise plays in them. It provides a theoretical basis for exercise to promote health and provides a potential mechanism for the correlation between muscle factor expression and immunity, as well as the involvement of exercise in body immunity. It also provides the possibility to find a suitable exercise training program for immune system diseases.

## 1. Introduction

Exercise is planned, structured, repetitive, and purposeful physical activity that aims to improve or maintain physical fitness [1]. Exercise has been shown to be able to participate in immune system regulation by influencing the function of various types of leukocytes and affecting a variety of physiological processes [2]. The influence of exercise on immune function has two sides. Although studies have shown that excessive exercise weakens the immune system, it is widely accepted that exercise of appropriate intensity benefits the body’s immune defences [3,4]. Therefore, exercise is receiving increasing attention as a new therapeutic strategy, and its molecular mechanisms for modulating the immune system deserve more in-depth study.

Exercise activates skeletal muscle as an endocrine organ and induces the production and secretion of small proteins (5–20 kda) and proteoglycan peptides by contraction and elongation of skeletal muscle. These small proteins and proteoglycan peptides are called myokines [5,6]. Irisin, interleukin-6/10/15 (IL-6, IL-10, IL-15), brain-derived neurotrophic factor (BDNF), fibroblast growth factors 2 and 21 (FGF2/21), leukemia inhibitory factor (LIF) and insulin-like growth factor 1 (IGF-1) and other myokines [5,7,8] have been proved to be able to resist chronic and inflammatory diseases, such as diabetes and tumor growth [9]. They also have a benign effect on the body. The above myokines have been shown to regulate the function of immune cells and improve their proliferation and differentiation ability, thus enhancing the immune function of the body and promoting its health (Figure 1). There is no article that systematically addresses the relationship between exercise, myokines, and immune cells. The aim of this review is to summarise the positive effects of exercise-induced myokines on immune cells, to discuss the important role played by exercise, to understand the intrinsic mechanisms by which the relevant myokines are involved in the immune regulation of the organism, and to provide possibilities for the screening of suitable exercise training protocols.

## 2. Potential Immune Cells Impacted by Exercise

Exercise can have a profound effect on the immune system, and can indirectly affect chronic diseases such as cancer, nonalcoholic fatty liver disease, and cardiovascular diseases through immune response [10,11]. Exercise regulates the function of immune cells, thereby improving the body’s immunity, which is an important way for exercise to affect the body’s function.

Exercise training can affect the body’s innate and adaptive immune processes. Innate immunity is the body’s first line of defense against invading pathogens. Cells involved in innate immunity include monocytes/macrophages, neutrophils, natural killer cells, and dendritic cells [12]. Macrophages are an important component of innate immunity and play an integral role in development, homeostasis, and host defense [13,14]. Macrophage polarization is the process by which macrophages functionally respond to microenvironmental stimuli and signals encountered in each specific tissue [15]. After being stimulated by different tissue sources or environmental stimuli, macrophages can be activated into classically activated macrophages (M1 type) and alternatively activated macrophages (M2 type) [16]. M1-type macrophages are usually induced by recognition by Th1 cytokines such as IFN-γ and TNF-α or by bacterial lipopolysaccharide (LPS). These macrophages produced and secreted high levels of the proinflammatory cytokines TNF-α, IL-1α, IL-1β, IL-6, IL-12, IL-23, and cyclooxygenase-2 (COX-2) and low levels of IL-10. Functionally, M1 macrophages participate in pathogen clearance during infection by activating the reduced nicotinamide adenine dinucleotide phosphate (NADPH) oxidase system and the subsequent generation of reactive oxygen species (ROS) [17,18,19]. M2 macrophages are anti-inflammatory and are polarized by Th2 cytokines IL-4 and IL-13 through the activation of STAT6 by the IL-4 receptor α (IL-4Rα). In addition to IL-4 and IL-13, other cytokines such as IL-10 can regulate M2 polarization by activating STAT3 via the IL-10 receptor (IL-10R) [20,21,22]. M2 macrophages have a strong phagocytic capacity, remove debris and apoptotic cells, promote tissue repair and wound healing, and have proangiogenic and profibrotic properties [15,22,23]. The balanced polarization of M1/M2-type macrophages plays a key role in organismal inflammation and injury. Regular exercise training has been reported to reduce macrophage infiltration into other sites of chronic inflammation [24]. In contrast, acute exercise has a strong stimulator effect on the phagocytosis, antitumor activity, and reactive oxygen species of M1 and M2 macrophages [25,26,27]. Neutrophils are another major branch of the innate immune system [28] that are the first cells to reach the site of inflammation and regulate T and B cells. After stimulation with different conditions, neutrophils will exhibit different subpopulations. The N2 subgroup plays the role of anti-inflammatory [29] and plays an important role in a variety of diseases such as myocardial infarction. Prolonged endurance exercise (0.5–3 h) may increase the number of neutrophils by up to 5-fold [30], improving the body’s anti-inflammatory ability. Innate immunity also includes natural killer cells (NK cells). NK cells are a group of immune cells derived from the proliferation and differentiation of lymphoid stem cells and are widely distributed in various lymphoid and non-lymphoid tissues, such as the spleen, lymph nodes, liver, and lungs [31,32]. NK cells mediate cytotoxicity to exert immune effects and regulate other leukocyte subsets of the innate and adaptive immune system through the release of antitumor factors and chemokines [33]. After high-dose resistance exercise, the number of NK cells increases [34,35], improving the body’s anti-inflammatory ability. As an important regulator of innate and adaptive immune response, dendritic cells (DC) are derived from red marrow stem cells of bone marrow and the spleen and are widely distributed in the spleen, lymph nodes, and connective tissues after development. The ability to confine foreign antigens to major histocompatibility complex (MHC) and present antigens to T lymphocytes [36]. Periodized endurance training is able to modulate DC development and shift them towards a more mature state [37]. Mature DC can induce an immune response from T cells [38].

Adaptive immunity includes both humoral and cellular immunity. B cells play a pivotal role in humoral immunity. In humans, B cells originate from hematopoietic stem cells in the bone marrow [39] and differentiate into plasma cells and memory B cells when stimulated by antigens. The former can synthesize and secrete immunoglobulins (antibodies) and survive for a short time in the body, while the latter can persist in the body for a long period of time and respond rapidly when the body is re-infected with the same antigen. Both high-dose endurance exercise and low-dose resistance exercise can elevate the number of circulating B cells [40,41,42,43]. T-lymphocytes are the main effector cells in cellular immunity. T cells can be divided into several subpopulations according to their functions in the immune response. This article mainly deals with two subpopulations. One is the helper T cells (Th cells), also known as CD4+ cells, which can secrete cytokines and regulate or assist the immune response. The other subpopulation is cytotoxic T cells (Tc cells), also known as CD8+T cells, which can bind directly to antigens and play an immune role [44]. The proliferation of CD8+T cells and CD4+T cells can be detected immediately after high-dose endurance exercise [34,35,41].

In short, the regulation of immune cell function, as an important component, plays a crucial role in the occurrence of immune responses in the body. If the immune cells function abnormally, the body becomes susceptible to various diseases. A series of studies have observed that selecting a more effective exercise program according to different diseases or exercise needs (that is, tailored exercise) can regulate the function of immune cells and induce the differentiation, proliferation, and activation of immune cells, which are essential for immune activity and thus contribute to health improvement [3]. However, the specific mechanisms by which exercise regulates immune cell function are unknown. Understanding exercise-induced myokines and their relationship to the immune response, particularly their role in regulating immune cell function, provides insight into the many benefits of exercise.

## 3. Exercise and Myokines

Myokines are involved in the regulation of exercise on health. Studies have found that exercise-induced myokines can produce a series of metabolic changes, which contribute to the prevention and treatment of a variety of chronic diseases [45]. Therefore, myokines may be the physiological basis for regulating and protecting body functions through exercise training (Table 1 and Figure 2). The regulation of myokine secretion by exercise can be affected by the mode, intensity, and duration of exercise [8]. It is very important to understand the regulatory effects of exercise on myokines.

### 3.1. Irisin

Irisin, a “star” myokine discovered in 2012, is a cleaved and secreted fragment of fibronectin type III domain containing 5 (FNDC5) regulated by receptor gamma coactivator 1α (PGC1-α) in muscle [77]. The amino acid sequence of irisin is highly conserved in most mammalian species (including humans) [78] and has been shown to be expressed in almost all tissues and organs of eukaryotes, with particularly high expression in skeletal muscle. One study reported that the expression of FNDC5 was nearly 200-fold higher in muscle tissue than in adipocytes [79]. In a recent study, Leger et al. observed that FNDC5/irisin may be expressed only in fast muscle fibres (gastrocnemius) and not in slow muscle fibres (flounder muscle) [80], but the specific mechanism needs to be further analysed in the future. It has been shown that irisin is associated with the browning of white adipose tissue (WAT) and lipolysis capacity [81,82]. Irisin has important regulatory roles in promoting organismal health, including inhibiting the development of chronic diseases and modulating the function of immune cells [47].

Irisin is one of the most important myokines that is clearly induced by exercise. Moderate-intensity treadmill exercise increased irisin levels in *mice* gastrocnemius and soleus muscles by approximately 50% [46]. After eight weeks of swimming exercise, irisin levels were significantly increased in the skeletal muscle of *mice* [48]. Kang et al. in their study found that a high-fat diet resulted in a significant reduction in serum irisin levels in SD *rats*. Irisin expression was inhibited by obesity and may be associated with its induced adverse effects. Eight weeks of low-intensity swimming exercise increased serum irisin levels by approximately 39.6% compared to the high-fat diet-fed group [49], indicating that irisin has a positive significance in exercise regulation of obesity. Effects of exercise on irisin secretion are also related to different types of exercise. Aerobic, resistance, and vibration exercise, as well as electrical stimulation of skeletal muscle all up-regulated the expression of irisin in *mice* myocardium, with the up-regulation by resistance exercise being more significant [50]. Exercise-induced irisin secretion has also been reported in a number of clinical trials. Boström et al. noticed that the level of irisin in basal plasma increased after 10 weeks of regular exercise [51]. Another study noted that 12 weeks of appropriate resistance exercise significantly increased resting serum irisin levels in older women compared to no exercise and that periodic, sustained exercise-induced irisin improved muscle function in older women [83]. Irisin levels are not only regulated by prolonged exercise, but studies have shown that muscle and circulating irisin increase immediately after acute exercise [52]. Serum irisin levels increased 1.2-fold after acute exercise in both non-exercising healthy individuals and exercise-trained prediabetic individuals compared with pre-acute exercise [53]. The ambient temperature during exercise also has an effect on irisin secretion. McCormick et al. observed a more pronounced increase in serum irisin levels after 3 h of moderate-intensity exercise in a high-temperature environment of 32 °C compared to 16 °C ambient temperature [54]. Interestingly, when the ambient temperature is very low, such as during winter swimming, serum irisin levels in swimmers are significantly reduced after exercise [84]. Although it has been shown that irisin expression is up-regulated within the bloodstream in response to cold stimulation alone [85,86], higher temperatures may be more favourable for irisin expression under exercise-induced conditions. This may be related to differences in the tissue specificity of the irisin source and the mechanism of induction. In conclusion, the potential positive effects of irisin can be better played by selecting appropriate exercise types and environmental ambient according to different genders and diseases.

### 3.2. Interleukins

Members of the interleukin family are low molecular-weight proteins or glycoproteins [87]. Among these cytokines, IL-6 and leukemia inhibitory factor (LIF), which belong to the IL-6 family [88], as well as two other interleukins, IL-10 and IL-15, have been found to be released by the muscle and to exert anti-inflammatory effects, which have a beneficial effect on the organism. Studies have shown that their secretion is regulated by different kinds of exercise. IL-6 can be released from skeletal muscle and adipose tissue into the blood to exert endocrine effects. It is also known as lipomyokine [55] and has metabolic and anti-inflammatory effects. Northoff et al. noted [89] that IL-6 may be the major systemic cytokine after strenuous exercise and is associated with the acute phase response of hepatocyte metabolism after exercise. Their testing of serum from 17 marathon runners before and after exercise showed that IL-6 levels were significantly elevated after exercise. Northoff judged at the time that IL-6 was more pro-recovery overall than the classic pro-inflammatory effect. Further research shows that the increase of IL-6 after exercise is mainly attributed to the secretion of skeletal muscle [90,91,92]. IL-6 is thought to trigger an anti-inflammatory cascade, inducing the production of anti-inflammatory cytokines, such as IL-10 and IL-1ra (IL-1 receptor antagonists), and inhibiting the production of pro-inflammatory cytokines Il-β and tumor necrosis factor α (TNF-α) [93]. In addition, IL-6 plays a positive role in the regulation of muscle growth and development [94]. Acute muscle contraction induces IL-6 release from skeletal muscle and its release into the circulatory system. The increase in IL-6 levels was correlated with the type, duration, intensity, and amount of muscle involved in exercise [56]. Similar to irisin, IL-6 produced by skeletal muscle is involved in the modulation of chronic diseases such as obesity. Ahn et al. observed that obesity induced by a high-fat diet inhibits IL-6 expression in skeletal muscle. The expression of IL-6 was significantly increased in skeletal muscle of HFD-fed *rats* that underwent stair climbing resistance exercise and treadmill aerobic exercise for 12 weeks [57]. In addition, LIF, as a member of the IL-6 family, has been demonstrated to be released by resistance and endurance training [7]. Jia et al. observed that eight weeks of interval exercise training (IET) significantly increased the protein level of LIF in *mice* gastrocnemius muscle [58] and reversed gastrocnemius muscle atrophy. Broholm et al. found that subjects had a significant increase in LIF levels in their muscles after a round of high-resistance quadriceps training. However, exercise-induced LIF expression is increased only in skeletal muscle, but not in plasma [59]. This suggests that LIF is autocrine and paracrine released by myofibroblasts. Exercise-induced LIF is released from skeletal muscle and promotes the proliferation of myoblasts through autocrine paracrine secretion, which plays a beneficial role in improving muscle mass.

Similar to IL-6 and LIF, the expression of IL-10 and IL-15 of the interleukin family is induced by exercise and plays an important role as myokines in vivo. IL-10 has several anti-inflammatory effects. Shadan et al. observed in animal experiments that the serum level of IL-10 in diabetic *rats* increased after HIIT exercise, which inhibited the expression of pro-inflammatory factors such as TNF-α and TGF in kidney tissue and improved the pathological damage of kidney tissue in diseased *rats* [60]. It has also been observed in clinical trials that exercise increases the level of IL-10. After 12 weeks of moderate-intensity exercise, weight loss and increased levels of serum IL-10 were observed in obese populations, and the elevated levels were significantly correlated with the expression of lipocalin as well as a decrease in the inflammatory factor TNF-α [95]. Lira et al. found that increased levels of IL-10 occurred immediately after high-intensity interval training (HIIT) and steady-state training (SST), and reached a maximum at 60 min [61]. Increased IL-10 levels inhibited the expression of proinflammatory factors such as IL-1 and TNF-α induced by the NF-κB pathway, which effectively inhibited the progression of the inflammatory response [62]. IL-15 was also found to be up-regulated after exercise. Clinical data showed that levels of IL-15 in the blood were elevated after resistance exercise [96]. Immediately after bilateral leg resistance training exercise, serum IL-15 levels increased approximately 5.3-fold. Four hours after exercise, the expression of IL-15Rα mRNA and protein in skeletal muscle increased by about 2 times and 1.3 times respectively compared with that at rest [63]. It was found that serum IL-15 concentration in non-active women decreased to below baseline level after the first high-intensity circuit training (HICT) session. After 15 HICT sessions, the serum IL-15 concentration increased and insulin resistance was improved. The sudden excessive intensity caused a decrease in IL-15 levels [64]. However, the level of IL-15 increased after adaptation and showed beneficial effects on the body. IL-15 levels may increase with fitness and be related to exercise intensity. The level of IL-15 may also be related to the level of female hormones. Compared with postmenopausal obese women, 12 weeks of regular resistance exercise increased IL-15 levels and reduced body fat in premenopausal obese women [97]. In summary, there are differences in the effects of different exercise types on the secretion of interleukin family myokines, and the proper form of exercise is important for the regulation of IL-6, IL-10, IL-15, and LIF expression levels in vivo. So far, the induced effect of HIIT is supported by a relatively large number of studies.

### 3.3. Brain-Derived Neurotrophic Factor (BDNF)

Brain-derived neurotrophic factor (BDNF) is a protein encoded by the BDNF gene. BDNF is mainly composed of β-fold and random coiled-coil secondary structure, containing three disulfide bonds, and is a basic protein. BDNF is synthesized as a pro-brain-derived neurotrophic factor (pro-BDNF) [98].

Pro-BDNF is cleaved intra- and extracellularly to mature BDNF (mBDNF or traditionally BDNF) [99]. mBDNF selectively activates the Tropomyosin receptor kinase B (TrkB) receptor [100], leading to enhanced phosphorylation of TrkB, which is predominantly found in the brain, but also exists to a lesser extent in skeletal muscle [101], where phosphorylation activates the Ras-MAPK pathway and finally CREB at the serine site of the cAMP response element binding protein (CREB) [102]. CREB promotes neuronal cell survival and enhances synaptic plasticity and neurogenesis by increasing the expression of the *BDNF gene* and the anti-apoptotic protein gene BCL-2 [103]. BDNF can participate in the regulation of brain and skeletal muscle functions. BDNF is widely distributed in the central nervous system, peripheral nervous system, endocrine system, bone, and cartilage tissues. It is expressed in non-neurogenic tissues, including skeletal muscle [104], and can be transported from the periphery to the brain across the blood-brain barrier [105,106]. At the same time, BDNF secreted from the brain can also enter the peripheral blood through the blood-brain barrier, and it is positively correlated with the change in serum BDNF level [107].

BDNF has been shown to be induced by exercise, which actually contributes to the understanding of exercise’s facilitation of neurodevelopment. The latest study by Sebastian et al., 2024 showed that the levels of pro-BDNF and mBDNF in plasma significantly increased after aerobic exercise [108]. The expression of pro-BDNF was higher in muscle, especially in type I fibers, but the expression of mBDNF was not detected in muscle. It is speculated that the exercise-mediated increase in circulating mBDNF may be derived in part from pro-BDNF cleavage produced by skeletal muscle release and in part from nerves and other tissues. The potential release of BDNF from skeletal muscle during exercise may occur in the form of pro-BDNF rather than mBDNF [108]. Studies have found a link between cognition and skeletal muscle function. There is a correlation between the decline of cognitive function and the decrease of skeletal muscle mass in hemodialysis patients, and muscle-derived BDNF plays an important role in this process [65]. Exercise increases BDNF levels in the skeletal muscle, plasma, and brain and significantly improves brain function. In preclinical trials, exercise training increased BDNF levels in the cerebral cortex, striatum, and hippocampus of *mice*, inhibited neuroapoptotic pathways in the cerebral cortex, promoted the release of dopamine, and improved cognitive function in mice [66,67,68]. In a clinical trial, Håkansson et al. observed a significant increase in serum BDNF levels and improved cognitive function in healthy older adults after 35 min of physical activity [109]. Increased levels of BDNF not only have beneficial effects in healthy older adults but also play an important role in Parkinson’s patients. It has been reported that the level of BDNF is decreased in patients with Parkinson’s disease (PD), and the progression of symptoms is correlated with the level of BDNF [69]. The level of mBDNF increased by 12% in Parkinson’s patients after acute aerobic exercise, which enhanced neural plasticity and improved exercise capacity [110]. Interestingly, recent studies have found that the myokines irisin can modulate BDNF levels. Circulating irisin and hippocampal BDNF levels change with exercise intensity, peaking at moderate exercise intensity and thus positively affecting brain function. The results showed that irisin secreted from skeletal muscle into the circulation but not brain-derived irisin-induced hippocampal BDNF expression [80]. Unfortunately, this study was not able to examine changes in BDNF levels in muscle and serum. These studies have demonstrated that aerobic exercise of appropriate intensity induces up-regulation of BDNF levels, improves cognitive function, and plays a protective role in brain nerves. However, excessive exercise may also have the opposite effect on BDNF expression. A significant reduction in serum BDNF levels and a reduction in the volume of the right hippocampal subregion of the brain were also observed during a six-week incremental maximal exercise test (IMET) [111]. The effect of BDNF on brain nerves may be related to the mode and intensity of exercise.

### 3.4. Fibroblast Growth Factors (FGFs)

Fibroblast growth factors (FGFs) are a structurally related family of 22 molecules. FGFs bind to four high-affinity, ligand-dependent FGF receptor tyrosine kinase molecules (FGFR1-4) [112]. The activation of FGFR promotes cell differentiation, migration, and survival [113]. FGF2 and FGF21, members of the FGFs family, are expressed in skeletal muscle and can be secreted to act on other tissues, thereby acting as myokines.

FGF2 is abundant in homogenates of muscle tissue and can be secreted by myotubes cultured in vitro [114]. Muscle tissue has been reported to release FGF2 after injury and strenuous exercise [70]. Sustained endurance exercise promotes FGF2 expression more than other exercises. In a preclinical trial, six weeks of continuous endurance training (CET) resulted in significantly higher gene expression of FGF2 and attenuated myocardial fibrosis in old *rats* compared with high-intensity interval training [71]. In contrast, in a one-year randomised interference trial, postmenopausal women who were usually physically inactive but healthy until the age of 50–74 years were asked to perform 150 min or 300 min of aerobic exercise per week, respectively. The results showed that 300 min of intense aerobic exercise per week resulted in a more pronounced increase in serum FGF2 expression [115]. FGF2 expression is clearly regulated by the timing and type of exercise. Unlike FGF2, it has been shown that FGF21 is expressed at very low levels in skeletal muscle under normal conditions [116]. Exercise stimulates muscle to produce FGF21, which is released into the circulation, increasing serum FGF21 levels [72]. The release of FGF21 is regulated by multiple exercise modes. Resistance exercise affects FGF21 levels. After eight weeks of incremental resistance training, FGF21 levels in the flounder muscle of obese *mice* were significantly increased, improving muscle strength [73]. FGF21 levels were significantly increased in human serum as well as in *mice* plasma and muscle after a single acute exercise session [117,118]. A 2020 meta-analysis showed that circulating levels of FGF21 peaked 1 h after acute exercise in subjects performing resistance, endurance, combined exercise, and intermittent exercise [74]. Cuevas-Ramos et al. observed that after two weeks of routine exercise, the serum FGF21 concentration of the subjects increased significantly [119]. In these studies, FGF21 levels increased after exercise. The opposite result was found in a study of fatty liver disease. Serum FGF21 levels were decreased in elderly men with fatty liver after five weeks of endurance training and were positively correlated with liver fat content [120]. This may indicate that the action of FGF21 is related to the regulation of inflammation. In summary, FGF2 and FGF21 are regulated by aerobic and anaerobic exercise. A variety of exercise modes elicit muscle secretion of FGF2 and FGF21.

### 3.5. Insulin-like Growth Factor-1 (IGF-1)

Insulin-like growth factors (IGFs), including IGF-I and IGF-II, are evolutionarily conserved peptides related to the insulin structure. Mature IGF-I and IGF-II consist of A, B, C, and D domains. Homology of the A and B domains of insulin-like growth factor to insulin [75]. It is highly expressed in muscle and is a key growth factor for muscle growth and skeletal development, and is secreted by skeletal muscle through autocrine and paracrine secretion [7]. The role of IGF-1 includes promoting cell proliferation, differentiation, and survival.

Studies have reported that both resistance training and aerobic exercise can enhance the secretion of IGF-1 [75,120]. Matheny et al. observed that 16 weeks of resistance training increased IGF-1 levels in the quadriceps, calf, and foot muscles of liver IGF-1-deficient (LID) *mice*. The up-regulation of local IGF-1 may be involved in the compensatory muscle growth after resistance exercise [76]. Both resistance and aerobic training for eight weeks can increase the expression of IGF-1 in the serum of patients with sarcopenic obesity, and the effect of combined exercise on promoting IGF-1 secretion is more obvious than that of resistance and aerobic exercise alone [121]. At present, there are few related studies on the regulation of IGF-1 by exercise, and more exercise programs related to IGF-1 need to be further studied.

## 4. Myokines and Immune Cells

In recent years, more and more literature has reported that myokines are involved in the regulation of the body’s immune function. These myokines are capable of modulating various immune cells. They can improve the proliferation and differentiation abilities of immune cells, so as to enhance the body’s immunity and promote health. It is essential to comprehend how different myokines regulate distinct immune cells to fully understand their role in immunological processes.

### 4.1. Irisin

The regulatory effect of irisin in vivo is related to the activation of immune mechanisms. As mentioned above, irisin induced by exercise plays a regulatory role in obesity. Indeed, a decrease in irisin levels was observed in all diseases including obesity [49], cancer [122], atherosclerosis [123], and diabetes [124]. The progression of these diseases is closely linked to inflammatory and immune responses. At present, it is believed that irisin induced by exercise plays an active role in regulating the function of macrophages and influencing the inflammatory response and immune response.

Irisin affects macrophage function through the regulation of cell activity, phagocytosis, antioxidant capacity, polarization, and apoptosis. Most of the experiments were observed through in vitro studies. The mechanism by which irisin acts on macrophages is shown in Figure 3. Excessive accumulation of reactive oxygen species (ROS) causes oxidative damage to cellular macromolecules, leading to cell necrosis [125]. It was found that irisin regulates macrophage activity by reducing the level of ROS and promotes macrophage proliferation in a dose-dependent manner, increasing their phagocytic capacity [126]. Macrophages stimulated with LPS were treated with irisin. Irisin inhibited the LPS-induced elevation of mitochondrial ROS, effectively reduced the production of free radicals in macrophages, and significantly reduced the production of harmful H2O2. ROS can inhibit the nuclear translocation of nuclear factorerythroid 2 p45-related factor 2 (Nrf2), destroy the redox homeostasis in macrophages, and lead to the proinflammatory response of M1 macrophages [127,128,129]. Irisin induction resulted in increased expression of the intracellular antioxidant and anti-inflammatory factors Nrf2 and heme oxygenase (HO)-1. Meanwhile, the expression and release of mobility group box 1(HMGB1) were significantly decreased. HMGB1 is a nuclear DNA-binding protein that is responsible for the regulation of gene transcription, which contributes to macrophage reprogramming to an inflammatory M1-like phenotype [130]. Irisin regulated the expression of key factors in the antioxidant signaling pathway Nrf2/HO-1/HMGB1, inhibited macrophage differentiation to a pro-inflammatory M1-like phenotype, and enhanced the antioxidant mechanism of activated macrophages [131]. The study by Dong et al. also confirmed that irisin modulates the expression of phenotypic markers in macrophages and induces macrophage differentiation. Irisin decreased the levels of the M1-like macrophage marker CD86 in lipopolysaccharide-induced macrophages while increasing the levels of the M2-like macrophage markers CD163 and CD206, which stimulated macrophage polarization from M1 to M2 and produced anti-inflammatory effects [124]. Other studies have found that irisin differentiates M0 and M1 macrophages toward the M2 phenotype through the AMP-activated protein kinase (AMPK) pathway [132]. Ye et al. used small interfering RNA for AMPK-α to block AMPK activation. AMPK-α siRNA significantly inhibited irisin-induced macrophage phenotypic shift to M2. Thus, confirming that irisin-induced M2 polarization is associated with activation of AMPK [132].

In addition, irisin was able to inhibit macrophage apoptosis. Zheng et al. found that irisin inhibited oX-LDL-induced nuclear translocation of transcription factor 6 (ATF6) and reversed clathrin-induced up-regulation of C/EBP homologous protein (CHOP), protein kinase RNA-like ER kinase phosphorylation (p-PERK), and eukaryotic translation initiation factor 2α phosphorylation (p-elF2α). It also up-regulated the expression of apoptosis inhibitor BCL-2, which attenuated macrophage apoptosis in vitro [133]. Not only that, irisin also improved macrophage infiltration. In a *mice* model of atherosclerosis induced with nicotine, nicotine increased macrophage infiltration and aggravated atherosclerosis. In contrast, after treatment with irisin, irisin inhibited macrophage infiltration by activating PTEN via integrin αVβ5 receptor, thereby inhibiting PI3K and promoting the up-regulation of the cell cycle protein-dependent protein kinase inhibitor P27. Thus, irisin significantly ameliorated nicotine-induced macrophage infiltration and inhibited the progression of atherosclerosis via the integrin αVβ5/PI3K/P27 pathway [134]. In summary, irisin can enhance the function of immune cells, especially macrophages, which is beneficial for the recovery of many chronic diseases.

### 4.2. Interleukins

Interleukins play an essential role in transmitting messages, activating and regulating immune cells, and mediating the activation, proliferation, and differentiation of T, B, macrophage and NK cells, as well as inflammatory responses [87]. The relevant details are described in detail below (e.g., Figure 4).

#### 4.2.1. IL-6 Family

In the IL-6 family, IL-6 and LIF, as myokines, are induced by exercise and regulate the function of immune cells. IL-6 is highly expressed in exercising skeletal muscle cells [135]. Several synergistic interactions between it and immune cells in muscle tissue have been suggested as determinants in the regulation of muscle injury and inflammation [136,137], and it is responsible for inducing regeneration-promoting pathways and anti-inflammatory processes in skeletal muscle [94,138,139]. IL-6 can regulate the proliferation and differentiation of immune cells [140], thus having beneficial effects on the body.

Studies have found that the IL-6 signaling pathway plays an important role in the regulation of macrophage differentiation [141]. The presence of IL-6 increased the expression of the M2-like macrophage marker CD206, that is, IL-6 promoted the polarization of macrophages to the M2 phenotype [142,143]. Ding et al. confirmed that IL-6 can promote the polarization of M2 macrophages through the STAT3 pathway, and activated M2 macrophages have a feedback-promoting effect on the invasion and migration of trophoblast cells, which has a beneficial effect on the maternal and fetal microenvironment of pregnancy [144]. Other studies have shown that IL-6 can prevent the death of macrophages induced by Streptococcus pneumoniae and reduce inflammatory damage in the lung by inhibiting pyroptosis [145]. In conclusion, IL-6 exerted anti-inflammatory effects by promoting M2-like polarization of macrophages and inhibiting macrophage death.

IL-6 not only regulates macrophage function but is also involved in lymphocyte activation [146]. Valença et al. observed that the presence of IL-6 caused the translocation of STAT3 from CD4+T cells to mitochondria. This helps maintain mitochondrial calcium ions, and thus enhances CD4+T cell motility, better mediating the immune response [146]. Lipopolysaccharide (LPS) was administered intravenously to mice harboring immature CD4+T cells. Treatment with LPS induced an increase in IL-6 levels and increased the number of CD4+T cells [147]. In addition, IL-6 was found to promote T-cell differentiation [148,149]. IL-6 promoted the differentiation of CD8+T cells in a *mice* model of Brucella infection and played a key role in Brucella clearance [148]. In a mouse model of breast cancer, Liu et al. found that cryo-thermal therapy significantly increased IL-6 levels in tumor tissues, and IL-6 induced dendritic cell phenotypic maturation, which promoted T cell differentiation, inhibited tumor cell proliferation, and exerted anti-tumor effects [149]. The effect of IL-6 on macrophages, T-lymphocytes and other immune cells helps to regulate and promote the immune response, which enhances one’s own immunity and resistance and also improves a variety of inflammatory lesions caused by low immunity.

As a member of the IL-6 family, LIF has been found to promote macrophage differentiation. However, few studies have been conducted. Yu et al. treated bone marrow-derived macrophages (BMDM) and THP-1 cell-induced differentiated human macrophages with recombinant human LIF (RLIF). Significant up-regulation of marker genes of the M2 phenotype was observed in both types of macrophages. Moreover, treatment with STAT3 inhibitor STAT3i significantly reduced the mRNA levels of M2 marker genes in BMDM [150]. This suggests that similar to IL-6, LIF can play an anti-inflammatory role by stimulating the conversion of macrophages to M2 type through the STAT3 signaling pathway.

#### 4.2.2. IL-10

Similar to the two members of the IL-6 family, IL-10 exerts main anti-inflammatory effects as a myokine. In in vivo experiments, infused in myocardial infarction mice, IL-10 stimulated M2 macrophage polarization and acted as an inhibitor of inflammation [151]. IL-10 also plays a central role in regulating the switch of muscle macrophages from the M1 to M2 phenotype in damaged muscle in vivo. In response to IL-10, the expression of CD163 and Arg1, markers of M2 macrophages, increased and stimulated myoblast proliferation. The recovery of the damaged muscle plays an important regulatory role [152]. In in vitro experiments, IL-10 induced anti-inflammatory effects by activating the STAT3 signaling pathway to increase the mRNA and protein expression of macrophage apoptosis inhibitor (AIM) in bone marrow-derived macrophages in *mice* [153]. Piao et al. found that the addition of IL-10 up-regulated the expression of P50 and P65, subunits of NF-κB, in the nucleus of macrophages and promoted P65 phosphorylation. The subsequent addition of sarsasapogenin, an inhibitor of the NF-κB pathway, which inhibited the degradation of the NF-κB inhibitory protein IκB and P65 phosphorylation, effectively inhibited the activation of the NF-κB pathway [154]. Inhibitors of the NF-κB pathway attenuated the expression of the M2 marker CD163. These results suggest that the effect of IL-10 on the M2 polarization of macrophages is partly dependent on the NF-κB pathway [155]. Taken together, the anti-inflammatory effect of IL-10 is also achieved mainly by promoting macrophage polarization toward the M2 type.

IL-10 is a major B-cell stimulating factor that affects B-cell proliferation and differentiation [156]. In an in vitro study, in the presence of IL-10, B cells [157] in germinal centers (GCs) formed by B cells could differentiate successively into CD20+CD38-memory B cells, and then into CD20-CD38+ plasma cells, which secrete antibodies and perform adaptive immune functions [158]. In another study, human B cells cultured in vitro were found to proliferate in an IL-10-dependent manner and subsequently differentiate toward antibody-secreting cells [156]. The research of Karmtej et al. has provided further evidence. Immunoglobulin (IgA) is a key immunoglobulin in the respiratory tract and gastrointestinal tract. Abnormalities in B cell function can lead to IgA deficiency [159]. IL-10, when added to peripheral blood monocytes, induces IgA production by B cells from IgA deficient patients. This proves that IL-10 can induce B cell proliferation, Ig class switching, and antibody secretion [160,161]. Thus, IL-10 is able to participate in the activation of adaptive immune responses in the body by promoting the proliferation and differentiation of B cells.

The effects of IL-10 on T cells have also been reported in the literature. IL-10 can increase T-cell survival and enhance cellular immunity. In clinical trials, Wang et al. observed that patients with high serum IL-10 levels had significantly higher T cell activity compared to patients with low serum IL-10 levels [162]. In preclinical experiments, Nir Yoyev et al., in their study of experimental encephalomyelitis (EAE) disease, showed that IL-10 signaling in CD4+T cells can promote CD4+T cell survival, which enhances autoimmunity in the central nervous system (CNS) [163]. Thus, IL-10 exerts an immune effect by enhancing the survival of T cells. In summary, IL-10 mainly affects the functions of macrophages, B cells, and T cells, thus enhancing the immune function of the body and promoting health.

#### 4.2.3. IL-15

IL-15 is a pleiotropic myokine that plays a crucial role in the function of immune cells, including NK cells and T cells. IL-15 promotes the proliferation of natural killer cells. Judge et al. observed a significant up-regulation of the level of CD69, an activation marker for NK cells, in tumor-infiltrating lymphocytes treated with IL-15, effectively treating soft tissue sarcoma (STS) [164]. The same conclusion was reached by Zhao et al. They observed that after intraperitoneal injection of IL-15 in a septic *rat* model, IL-15 increased the number of NK cells in the peripheral blood of septic *rats* in a dose-dependent manner and prolonged the survival time of septic rats [165]. Molecular biology studies showed that IL-15 treatment increased the phosphorylation intensity of STAT5a/b in the JAK/STAT pathway and TOR, AMPKα1, and AKT(T308) in the PI3K/AKT pathway. TERT (telomerase) was activated, inducing the proliferation of NK cells. When JAK/STAT and PI3K/AKT pathway inhibitors were added, the expression of TERT in NK cells was significantly reduced [166]. This suggests that IL-15 could enhance the activity of NK cell TERT through both JAK/STAT and PI3K/AKT signaling pathways, thus significantly increasing the number of NK cells. In clinical experiments, synthetic IL-15 (rhIL-15) was administered intravenously to patients with advanced metastatic solid tumors by infusion or flow pump. The results showed that the injection of rhIL-15 had a greater effect on NK cell expansion. The injection of IL-15 enhanced the cytotoxic activity of NK cells, made them proliferate vigorously, and enhanced the anti-tumor effect [167].

On the other hand, IL-15 can regulate the proliferation and activation of T lymphocytes, especially CD8+T cells. Levels of the proliferation marker Ki67 were significantly up-regulated in CD8+T cells after IL-15 treatment [164]. Choi et al. observed an increase in the phosphorylation levels of mTOR and ribosomal protein S6 (a downstream molecule of mTOR) and promoted the proliferation of memory CD8+T cells after stimulating the memory CD8+T cell population with IL-15 [168]. In addition, IL-15 can be involved in the activation of CD8+T cells through the JAK/STAT signaling pathway. Chen et al. observed that H_2_O_2_ promoted the expression of IL-15 in keratinocytes and promoted the phosphorylation of STAT3 and STAT5 in memory CD8+T cells (CD8 T+EMS). The addition of JAK/STAT signaling pathway inhibitors significantly reduced keratinocyte-derived IL-15-induced phosphorylation of STAT3 and STAT5 in CD8 T+EMS. Thus, oxidative stress-induced IL-15 contributes to the activation of CD8 T+EMS through the JAK-STAT signaling pathway and exerts an immune effect [169]. In clinical experiments, researchers found that in the presence of IL-15, the proliferation of recipient bone marrow CD8+ T cells increased, and the number of total CD8+T cells in peripheral blood tended to increase and promote the activation of CD8+T cells [170]. IL-15 not only regulates the proliferation and activation of CD8+ T cells but also participates in the regulation of other T cell functions. Programmed cell surface death (PD-1) has been reported to inhibit the PI3K/AKT pathway, leading to T cell dysfunction [171]. Saito et al. treated aged septic mice with IL-15, which suppressed the expression of T cells and regulatory T cells (Treg) PD-1 and increased the proportion of naïve T cells and CD8+T cells, thereby ameliorating sepsis-induced T cell failure in aged septic mice [172]. In summary, IL-15 plays a crucial role in the proliferative function of NK cells and T cells.

### 4.3. BDNF

It has been reported that BDNF is involved in the regulation of immune responses, mainly by promoting macrophage polarization [173,174,175,176]. Sasaki et al. observed that after the treatment of *mice* macrophage cell line RAW264.7 cells with recombinant BDNF, BDNF promoted the expression of phosphorylated Rac1. Rac1 affects the depolymerization of cortical actin and increases cell migration, which enhances phagocytosis and reduces inflammatory stimuli in macrophages [173]. BDNF exerts anti-inflammatory effects by enhancing the phagocytic activity of macrophages through the Rac1 signaling pathway. In addition, BDNF can promote the transition of macrophage M1 to M2 phenotype. In one study, chronic BDNF vectors were injected into adult *C57 mice* undergoing T10 spinal cord injury (SCI). These animals exhibited a higher proportion of M2 phenotype macrophages compared to controls. This improves the inflammatory microenvironment, enhances the neuroprotective effect, and helps in the recovery of motor function after SCI [174]. Bi et al. observed increased serum BDNF levels and decreased STAT3 protein levels and phosphorylation in *mice* with diabetes mellitus-accelerated atherosclerosis (DMAS), whereas the overexpression of BDNF reduced the expression of markers of M1 macrophages and increased the expression of markers of M2 macrophages [175,176]. BDNF induces the differentiation of mouse macrophages to M2 type by inhibiting the STAT3 pathway, thus attenuating DMAS. It can be concluded from the above results that BDNF can achieve immunomodulation by enhancing the phagocytosis activity of macrophages and promoting differentiation to the M2 type.

### 4.4. FGF Family

The fibroblast growth factor (FGF) family is a group of pleiotropic growth factors that play important roles in cell proliferation, differentiation, angiogenesis, tissue repair, and regeneration [177]. FGF activates various downstream signaling pathways such as AMPK, NF-κB, etc. [178]. Among them, FGF2 and FGF21 are secreted by skeletal muscle and play the role of myokines. They mainly mediate macrophage polarization and play an immunomodulatory role in various diseases such as tumors and chronic pancreatitis.

#### 4.4.1. FGF2

FGF2 is an important regulator of macrophage differentiation [179]. Im et al. observed that tumor-associated macrophages in tumors of FGF2 gene-deficient (Fgf2LMW-/-) mice were biased toward an inflammatory (M1) phenotype. After treatment with exogenous FGF2, FGF2 altered the phenotype of tumor-associated macrophages [179], that is, transformed tumor-associated macrophages into an M2-like phenotype in the tumor microenvironment. Similarly, in studies of gastric cancer, FGF2 was found to be positively correlated with macrophage infiltration. After co-culturing HGC-27, a GC cell line with a relatively high expression of FGF2, with THP-1 cells, FGF2 secreted by HGC-27 inhibited the polarization of M1 macrophages and promoted macrophage polarization toward the M2 type [180] that is involved in the immunomodulation of the organism.

#### 4.4.2. FGF21

FGF21 is an important regulator of muscle growth, inflammation, systemic metabolism and premature aging [79]. It has been found that FGF21 ameliorates the inflammatory state of macrophages [181]. In one study, recombinant human fibroblast growth factor 21 (rhFGF21) was used to treat middle cerebral artery occlusion (MCAO) *mice*. Compared with the control group, the number of CD68+ and CD86+ macrophages in the rhFGF21 treatment group was significantly reduced, and rhFGF21 inhibited the transformation of macrophages to M1 phenotype and played an anti-inflammatory role. It can promote the functional recovery of stroke *rats* [182]. FGF21 can regulate macrophage polarization through the m-TOR signaling pathway. Wang et al. observed a significant reduction in the expression of both the M1 marker iNOS and the M2 marker CD206 in macrophages from mice with chronic pancreatitis (CP) treated with FGF21. This process can be blocked by m-TOR inhibitors. Thus, FGF21 ameliorates pancreatic fibrosis in chronic pancreatitis by reducing M1- and M2-type macrophages in CP mice via the m-TOR signaling pathway [183]. FGF21 also promotes macrophage polarization through the AMPK and NF-κB pathways. Kang et al. found that FGF21 activates hepatic AMPKα and reduces the phosphorylation level of NF-κB in the liver tissue of senescent mice. Further studies using in vitro experiments showed that FGF21 induced endotoxin-triggered phosphorylation of AMPKα and inhibited p-NF-κB/NF-κB levels in macrophages and promoted M1 to M2 polarization [184]. Therefore, it can be concluded that FGF21 can promote macrophage polarization through AMPK and NF-κB pathways, increase the expression of M2 macrophages, decrease the expression of M1 macrophages, and produce an anti-inflammatory effect, thus attenuating hepatic senescence injury. FGF21 can promote macrophage M2-type polarization through different signaling pathways and play an important regulatory role in chronic diseases such as inflammation.

### 4.5. IGF-1

The growth-promoting endocrine hormone IGF-1, which acts as a myokine, has less reported mechanisms for regulating immune cells. It has been reported in the literature that IGF-1 regulates neutrophil differentiation. Nederlof et al. observed that in vitro and in acute myocardial infarction (AMI) patients, IGF-1 mediated neutrophil polarization to an N2 phenotype and observed phosphorylation of STAT6. Subsequent addition of a JAK2 inhibitor markedly inhibited the up-regulation of N2 markers and the phosphorylation of STAT6 disappeared. It can be concluded that IGF-1 is able to improve the prognosis after myocardial infarction by enabling bone marrow-derived neutrophils to develop an anti-inflammatory N2 phenotype through the atypical signaling pathway, the JAK2-STAT6 pathway [185]. IGF-1 can differentiate neutrophils to produce anti-inflammatory effects. Currently, there are fewer studies on IGF-1, and more experiments are needed for more in-depth studies in the future.

## 5. Discussion and Conclusions

Exercise-induced myokines have been shown to be involved in the regulation of immune cell function, influencing the body’s innate and adaptive immune processes. This article reviews and discusses the effects of myokines on immune cell function and the important role that exercise plays in it. Exercise-induced myokines can affect the proliferation, differentiation, and survival of a wide range of immune cells, and a variety of exercise types, intensities, and durations can have an effect on myokine secretion. At present, there is no clear criterion for classifying exercise intensity. We referred to the criteria proposed by the American College of Sports Medicine (ACSM) [186] (Appendix A). Available evidence suggests that aerobic exercises, such as low to moderate-intensity swimming and running, endurance exercise, etc., and anaerobic exercise, such as resistance and strength, can promote the secretion of myokines by skeletal muscles. These myokines can enhance the function of immune cells, thereby improving the body’s immunity and promoting good health.

However, there is some uncertainty in current research on the link between exercise, myokines, and immune cells. As we mentioned at the beginning of the article, excessive exercise may also lead to the weakening of immune function [4]. In the future, the sports activity plan based on the standard prescription principle (such as the FITT principle [187]) will help to standardize the appropriate frequency, intensity, time, and type of exercise, and perhaps better play the role of exercise in promoting health. It should be noted that exercise-induced myokines are not always positive for the immune system. For example, follistatin-like protein 1 (FSTL1) and decorin (DCN), which are induced during exercise and secreted by muscles, can promote M1-like polarization of macrophages, which is considered a sign of promoting the progress of related diseases [188,189,190]. Among the myokines reviewed in this paper, interleukins can not only play an immunomodulatory role as myokines but also be widely recognized as inflammatory cytokines. Their role in immune regulation is still controversial. The different effects of these myokines may provide new perspectives on the molecular basis of exercise intervention in immunity. Our understanding of motor regulation of the myokines is not deep enough to fully explain the specific mechanism of the action of myokines on immune cells. In addition, beyond the effects of exercise, the secretion of myokines may also be related to factors such as diet and aging [191,192]. A large number of experiments are needed to study related issues in the future.

Nevertheless, our article reviews the regulation of immune cells by exercise-induced myokines, providing evidence for the critical role of exercise-induced myokines in enhancing the body’s immunity. Investigating the effects of induced myokines on immune cell function under different exercise modes, dosages, and intensities will provide a theoretical basis for further optimizing exercise health programs and improving exercise prescriptions for patients with chronic diseases. It is expected that the study of exercise-induced myokines will help us to further understand and explain the beneficial effects of exercise in the maintenance of health.

## Figures and Tables

**Figure 1 biomolecules-14-01205-f001:**
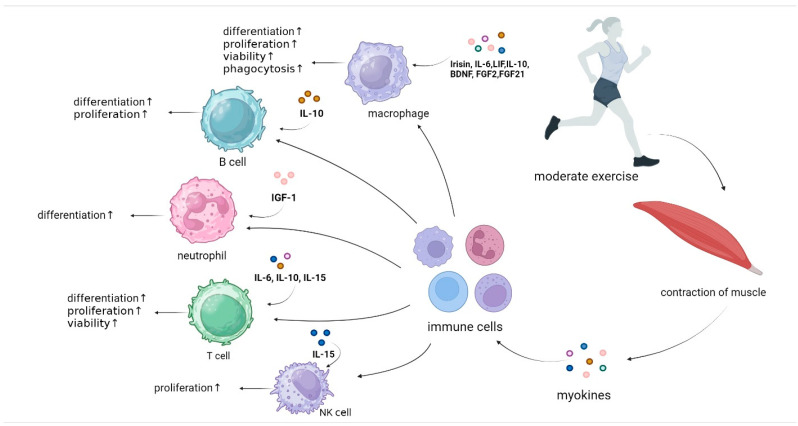
Exercise-induced myokines are involved in the regulation of immune cell function|Exercise induces contraction of skeletal muscle, which promotes the secretion of myokines, including irisin, IL-6, LIF, IL-10, IL-15, BDNF, FGF2, FGF21, and IGF-1. They can act on the immune system, affecting the processes of innate and adaptive immunity and regulating the function of immune cells. ↑: up-regulation (Created with BioRender.com).

**Figure 2 biomolecules-14-01205-f002:**
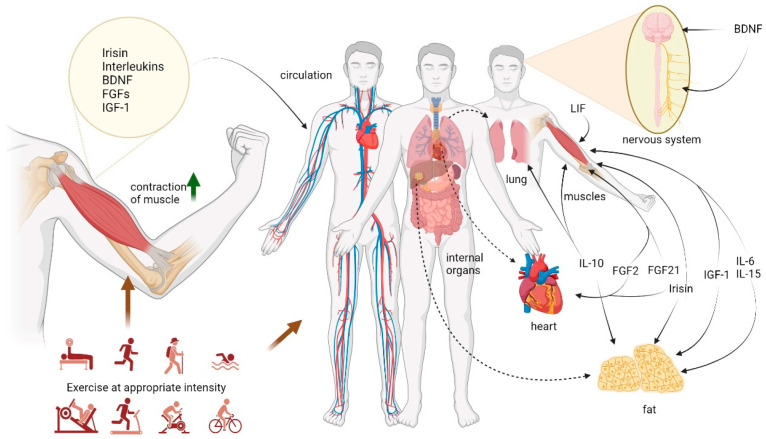
Exercise-induced myokines positively regulate the circulatory system, heart, lung, muscle, and nervous system|A variety of exercises induce the release of myokines from skeletal muscles. Myokines act on nerve, muscle, heart, lung, fat, and other tissues to promote health. (Created with BioRender.com).

**Figure 3 biomolecules-14-01205-f003:**
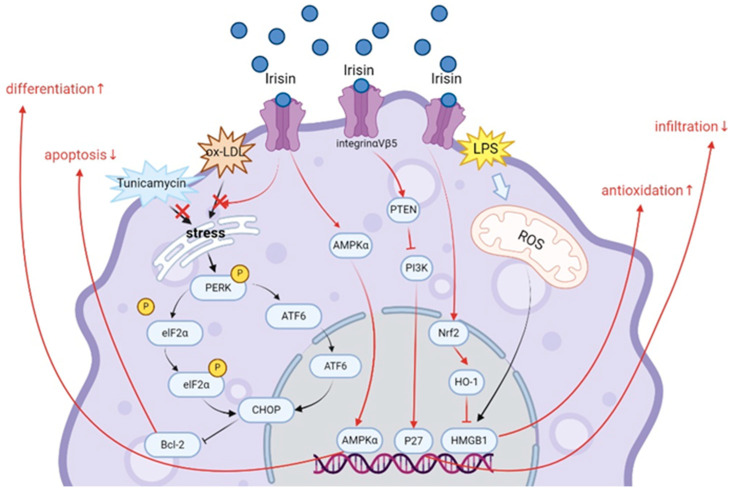
Irisin Signaling Pathways in Macrophages|Irisin inhibits apoptosis, promotes differentiation, reduces infiltration, and enhances the antioxidant capacity of macrophages via the PERK/eIF2α/CHOP and ATF6/CHOP endoplasmic reticulum stress signaling pathways, the AMPK pathway, the integrin αVβ5/PI3K/P27 pathway, and the Nrf2/HO-1/HMGB1 pathway. ↑: up-regulation; ↓: down-regulation (Created with BioRender.com).

**Figure 4 biomolecules-14-01205-f004:**
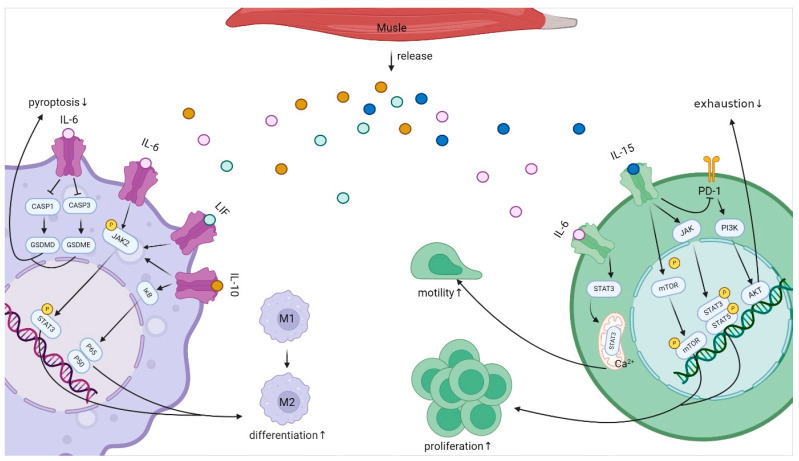
Interleukin Signaling Pathways in Macrophages and T Cells|IL-6 reduces macrophage pyroptosis, promotes their differentiation, and enhances T-cell motility through the Caspase-3-GSDME pathway, Caspase-1-GSDMD pathway, JAK/STAT3 pathway, and STAT3/mitochondrial pathway; LIF promotes macrophage differentiation through the JAK/STAT3 pathway; IL-10 promotes T-cell differentiation and reduces T-cell motility through the JAK/STAT3 pathway; IL-15 promotes T cell differentiation and reduces T cell exhaustion via the mTOR pathway, JAK/STAT pathway, and PI3K/AKT pathway.↑: up-regulation;↓: down-regulation (Created with BioRender.com).

**Table 1 biomolecules-14-01205-t001:** Changes in myokines after exercise.

Myokines	Model	Stimulation	Myokine Alteration	Physical Signs	Ref.
Irisin	*Mice*	Swimming training (30 min/day, for 3 days/week for 8 weeks)	Irisin ↑ in muscle	Reduced the oxidative stress index (OSI), degeneration in the heart muscle, inflammation and cardiopathy	[46]
	*Mice*	Treadmill exercise with moderate intensity (5 days/week for 8 weeks)	Irisin ↑ in muscle	Increased bone mineral density of trabecular bone in mice	[47]
	*Rat*	Swimming training (60 min/day, for 7 days/week for 8 weeks)	Irisin ↑ in serum	Effectively improved bone health caused by obesity	[48]
	*Mice*	Aerobic exercise, Resistance exercise, Vibration exercise, The Electrical stimulation (60 min/day, for 5 days/week for 4 weeks)	Irisin/FNDC5 ↑ in muscle	Promoted mitochondrial autophagy, improving heart function and resisting exercise.	[49]
	*Human*	Cycling on stationary bikes (20–35 min/week for 10 weeks)	Irisin ↑ in plasma	Improved glucose homeostasis and caused a small weight loss	[50]
	*Human*	Progressive resistance training (1 h/day, for 2 days/week for 12 weeks)	Irisin ↑ in serum	Increased grip strengthand leg strength	[51]
	*Human*	Strength and endurance training intervention (60 min/day, for 2 days/week for 12 weeks)	Irisin ↑ in serum	Not described	[52]
	*Human*	Moderate-intensity treadmill walking (180 min, 21.9 °C and 41.1 °C)	Irisin ↑ in serum	Reduced oxidative stress and inflammation	[53]
	*Human*	Winter swimming	Irisin ↓ in serum	Not described	[54]
IL-6	*Human*	Marathon	IL-6 ↑ in plasma	Not described	[55]
	*Rat*	Treadmill running and ladder climbing (75 min/day, for 3 days/week for 12 weeks)	IL-6 ↑ in muscle	Helped reduce inflammation	[56]
LIF	*Rat*	Interval exercise training (60 min/day, for 5 days/week for 8 weeks)	LIF ↑ in muscle	Reduced apoptosis and promoted proliferation in gastrocnemius muscle	[57]
	Human	Heavy resistance exercise of 6–8 repetitions	LIF ↑ in muscle	Not described	[58]
IL-10	*Rat*	HIIT (Running with 8 m/min, 10 min/day for 5 days)	IL-10 ↑ in serum	The expression of pro-inflammatory factors was inhibited	[59]
	*Human*	Physical activity of moderate intensity (12 weeks)	IL-10 ↑ in serum	Improvement of metabolic risk factors	[60]
	*Human*	Running 5 km intermittently; running 5 km continuously at 70% of MAS (determined in the incremental test) on the treadmill	IL-10 ↑ in serum	It effectively inhibits the progression of inflammatory response	[61]
IL-15	*Human*	A bilateral leg resistance exercise	IL-15 ↑ in muscle serum	Improved myofibrillar fractional synthetic rate	[62]
	*Human*	High-intensity circuit training (HICT) (3 days/week for 5 weeks)	IL-15 ↑ in serum	Improved an insulin sensitivity	[63]
	*Human*	Moderate intensity exercise (60 min/day, for 3 days/week for 12 weeks)	IL-15 ↑ in serum	Decreased body fat	[64]
BDNF	*Mice*	Swimming training (30 min/day, for 7 days/week for 12 weeks)	BDNF ↑ in cerebral cortex	Provided neuroprotective effects	[65]
	*Mice*	Voluntary wheel-running exercise (30 days)	BDNF ↑ in striatum	Enhanced striatal dopamine (DA) release	[66]
	*Mice*	Rat cages equipped with running Wheels (3 h/day for 2 weeks)	BDNF ↑ in hippocampal tissues	Improved cognition in Alzheimer’s disease (AD)	[67]
	*Human*	Aerobic exercise (35 min)	BDNF ↑ in serum	Improved cognition in Alzheimer’s disease (AD)	[68]
	*Human*	Walk on a treadmill at light to moderate intensity (30 min)	mBDNF ↑ in serum	Enhancement of neuroplasticity and facilitate the improvement of motor performance	[69]
FGF2	*Rat*	Continuous exercise training (15 min at 65% maximal speed for 1 week, 20 min for 2 weeks, 25 min for three weeks at 70% maximal speed, and 30 min for 4, 5, and 6 weeks at 70% maximal speed)	FGF2↑ in heart tissue	Delayed age-related myocardial fibrosis	[70]
	*Human*	Aerobic exercise (300 min/week for 12 months)	FGF2 ↑ in serum	Reduced postmenopausal breast cancer risk	[71]
FGF21	*Mice*	Resistance training (10 repetitions/day, 3 days/week for 8 weeks	FGF21 ↑ in muscle	Improved muscle strength	[72]
	*Mice*	Performed treadmill exercises at 30 m/min for 60 min	FGF21 ↑ in plasma and muscle	Not described	[73]
	*Human*	A treadmill exercise test (following the Bruce’s protocol) (5 days/week for 2 weeks)	FGF21 ↑ in serum	Increased glucose intake	[74]
IGF-1	*Mice*	Ladder climbing (85-degree incline, 1.5 cm spacing), utilizing progressive overload (twice a day, every third day for 16–18 weeks)	IGF-1 ↑ in muscle	Compensatory growth of muscle	[75]
	*Human*	Resistance training (RT), aerobic training (AT), combination training (CT) (60 min/day, for 3 days/week for 8 weeks)	IGF-1 ↑ in serum	Increased muscle mass and reduced total fat mass and visceral fat area (VFA)	[76]

↑: up-regulation; ↓: down-regulation.

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
