# Peer review of "Myokines May Be the Answer to the Beneficial Immunomodulation of Tailored Exercise—A Narrative Review"

_biomolecules, 2024, doi:10.3390/biom14101205_

Round 1

Reviewer 1 Report

Comments and Suggestions for Authors

The review entitled “Myokines may be the answer to the beneficial immunomodulation of moderate exercise-a systematic review” by Zheng Lu et al. addresses a topic of pivotal importance  in athletes and general population health. Although the review is quite well organized and written, there are some concerns.

-          Title: the term “moderate” is quite misleading. Moderate exercise is defined as exercise at first ventilatory threshold or first lactate threshold or exercise between 3-6 METs (doi: 10.3389/fphys.2021.682233). I would suggest changing “moderate” in “tailored” and add a definition.

In this context,  specifying the difference between “exercise” and “physical activity would be worth (Caspersen et al., 1985).

-          Define the acronym IL in parentheses after “interleukin”, Introduction, line 37.

-          Insert comma after “diseases” , Introduction, line 36.

-          In paragraph 2 dendritic cells are missing and should be added in the list of  innate immune cells, line 61. Furthermore, dendritic cells are mentioned related to IL-6, line 476.

-          Please address the possible consequences of the increase in neutrophils and NK cells induced by prolonged endurance exercise and high-dose resistance exercise, lines 90 and 97, respectively, paragraph 2.

-          The sentence “T-lymphocytes are the main effector cells in cellular immunity”, line 104, should be moved to line 106, before the sentence “T cells can be divided into several…”, for consistency.

-          The sentence “thus contributes to the improvement of the health of the body”,  line 118, should be better addressed, as the beneficial effects of exercise strictly depend on exercise dose (please see doi.org/10.3389/fimmu.2022.954994).

-          I suggest changing the title of chapter 3, since “sport” is not just running or cycling but it is defined as individual or group activity in which participants chase a goal (Khan et al., 2012 Lancet). Exercise, i.e., would fit better in the context.

Furthermore, a short introduction on myokines is missing; please define them and specify their specific temporal dynamics.

-          Please insert a reference after the sentence “The regulation of myokines secretion by exercise can be affected by the mode, intensity, and duration of exercise”, line 128.

-          Concerning Irisin regulation and temperatures, lines 180-181, data by Polyzos et al., 2017 and other studies assert that low temperatures upregulate this myokine leading to browning of adipose tissue; please address this topic.

-          mBDNF (not MBDNF) even after a full stop, line 258 line; otherwise, change in Mature BDNF.

-          I would suggest addressing ROS level in relationship to immune system (i.e., ROS accumulation could be harmful for immune system through Nrf2, Wang et al., 2010; Hieltqvist et al., 2007).

-          The Authors assess that this is a systematic review (even if it seems quite narrative) thus Methods should be included.

-          There are several misspelling and grammar should be revised through the text (i.e., the last sentence of figure 2 legend misses the subject).

-          Acronyms should be defined the first time in the text and indicated in parentheses after extended definition, instead there are many acronyms without definitions (i.e., HICT, as high intensity circuit training, 240 line; H0-1, line 389 line; AMPK, line 403, and so on).

Comments on the Quality of English Language

English should be revised

Author Response

1.Summary

Thank you very much for your comments, which has brought great inspiration to us. Based on your suggestions, we have tried our best to make adjustments and believe it will bring great improvements to our articles. The adjustments are as follows.

2.Point-by-point response to Comments and Suggestions for Authors

Comments 1: Title: the term “moderate” is quite misleading. Moderate exercise is defined as exercise at first ventilatory threshold or first lactate threshold or exercise between 3-6 METs (doi: 10.3389/fphys.2021.682233). I would suggest changing “moderate” in “tailored” and add a definition.

In this context,  specifying the difference between “exercise” and “physical activity would be worth (Caspersen et al., 1985).

Response 1: Thank you for your valuable comments. After we carefully read this literature you provided, we did find that "moderate" is quite misleading. Change "moderate" to "tailored" as you suggested and define it on page 3, we add “selecting a more effective exercise program according to different diseases or exercise needs”lines 125-127. In addition, the definition of "exercise" was added in lines 25-26 on page 1 of the introduction section :” Exercise is planned, structured, repetitive, and purposeful physical activity that aims to improve or maintain physical fitness”.

Comments 2: Define the acronym IL in parentheses after “interleukin”, Introduction, line 37.

Response 2: Thank you very much for your valuable advice. We have added "interleukin" in line 37.

Comments 3:  Insert comma after “diseases” , Introduction, line 36.

Response 3: Thank you very much for your careful advice. We insert comma after "diseases," now on line 40.

Comments 4: In paragraph 2 dendritic cells are missing and should be added in the list of  innate immune cells, line 61. Furthermore, dendritic cells are mentioned related to IL-6, line 476.

Response 4: Thank you very much for your careful advice. This is really an oversight on our part. Now, we have added the dendritic cells in line 64, paragraph 2, and the description of the dendritic cells and the effects of exercise on the dendritic cells have been added in lines 100-107 :” As an important regulator of innate and adaptive immune response, dendritic cells (DC) are derived from red marrow stem cells of bone marrow and spleen and are widely distributed in spleen, lymph nodes and connective tissues after development. The ability to confine foreign antigens to major histocom-patibility complex(MHC) and present antigens to T lymphocytes[36]. Periodized endurance training is able to modu-late DC development and shift them towards a more mature state[37]. Mature DC can induce an immune response from T cells[38].”

Comments 5: Please address the possible consequences of the increase in neutrophils and NK cells induced by prolonged endurance exercise and high-dose resistance exercise, lines 90 and 97, respectively, paragraph 2.

Response 5: Thank you very much for your advice. According to your suggestion, we have reviewed the relevant literature. The possible consequences of the increase in neutrophils and NK cells induced by prolonged endurance and high-dose resistance exercise are” improving the body's anti-inflammatory ability” are added in lines 93 and 100, paragraph 2.

Comments 6: The sentence “T-lymphocytes are the main effector cells in cellular immunity”, line 104, should be moved to line 106, before the sentence “T cells can be divided into several…”, for consistency.

Response 6: Thank you for your valuable advice. We have adjusted the sentence order, now in lines 115-118. Your suggestions make our article more smooth.

Comments 7: The sentence “thus contributes to the improvement of the health of the body”,  line 118, should be better addressed, as the beneficial effects of exercise strictly depend on exercise dose (please see doi.org/10.3389/fimmu.2022.954994).

Response 7: Thank you very much for the literature. We read this reference carefully and adjusted lines 193 to 196. We wrote that “the potential positive effects of irisin can be better played by selecting appropriate exercise types and ambient temperature according to different genders and diseases” in 3.1. The article you provided is very valuable for reference. We decided to quote it on line 31 ”Although studies have shown that excessive exercise weakens the immune system, it is widely accepted that exercise of appropriate intensity benefits the body's immune defences”.

Comments 8:  I suggest changing the title of chapter 3, since “sport” is not just running or cycling but it is defined as individual or group activity in which participants chase a goal (Khan et al., 2012 Lancet). Exercise, i.e., would fit better in the context.

Furthermore, a short introduction on myokines is missing; please define them and specify their specific temporal dynamics.

Response 8: This is an excellent suggestion. Based on your suggestion, we believe that "exercise" is better.

 In addition, we added the introduction of myokines (these small biologically active proteins and proteoglycan peptides are known myokines) in lines 34 to 36 of the introduction.

Different myokines may be affected by different factors, including diet, exercise and so on. For exercise-induced myokines, there are many different results due to the influence of exercise form, time, period and other factors. There are also differences in the temporal specificity of myokines expression in different human studies. However, most studies have recognized that one-time and long-term exercise induced the expression of these myokines. In the part of " exercise and myokines ", we discuss the previous researches and hope to give readers a reference.

Comments 9: Please insert a reference after the sentence “The regulation of myokines secretion by exercise can be affected by the mode, intensity, and duration of exercise”, line 128.

Response 9: Thank you very much for your careful advice. We have added references at line 140 [8] (Domin, R.; Dadej, D.; Pytka, M.; Zybek-Kocik, A.; Ruchała, M.; Guzik, P. Effect of Various Exercise Regimens on Selected Exercise-Induced Cytokines in Healthy People. Int J Environ Res Public Health. 2021, 18.

Comments 10: Concerning Irisin regulation and temperatures, lines 180-181, data by Polyzos et al., 2017 and other studies assert that low temperatures upregulate this myokine leading to browning of adipose tissue; please address this topic.

Response 10: Dear reviewer, you have raised a very good question. It is undeniable that there is still a lot of controversy about irisin research so far. Therefore, we have carefully studied the literature you have presented (Polyzos et al., 2017 ), as well as other relevant literature, in the hope of answering your query properly.

We note that Polyzos et al. state in their paper that‘Once released into the circulation during exercise or cold exposure, irisin stimulates UCP1 expression and (white adipose tissue) WAT browning, increasing total body energy expenditure by increasing UCP1-mediated thermogenesis.’This view is largely based on their literature16 (Zhang et al., 2017).(Literature 15,Lee et al.,2014).

Zhang's related ideas, on the other hand, come from his knowledge of studies such as Lee et al.(The literature 15 in the Zhang et al. study,Lee et al.,2014).

Lee et al. explored the induced secretion of human irisin in response to cold exposure by observing changes in plasma irisin concentrations in unexercised (stable body core temperature) healthy adults in response to cold stimuli and comparing them to plasma irisin concentrations in normothermic, exercising healthy adults. It was found that the induction of irisin secretion was proportional to the intensity of human shivering, and its magnitude was similar to that secreted in response to exercise stimuli. Related studies are illuminating for understanding the role of irisin in fat milk browning and its link to lipolysis for heat production, but their experimental scenario design is markedly different from the more complex exercise state of winter swimming.

On the other hand, as we mentioned in the text, irisin is expressed in both muscle and adipose tissue (3.1 irisinline153-164). These different sources of irisin may exhibit completely different properties. As Zhang et al. point out in their researchers ‘Different laboratories have tried to study the activation of PGC1α/FNDC5/irisin signalling by exercise in skeletal muscle or WAT, and they have obtained results that are difficult to reconcile.’

Taken together, we suggest that observations of irisin expression in different tissue sources as well as experimental designs may have led to different conclusions. There must be a difference between exercise induction and cold stimulation alone on the induction of irisin. In this paper, a synthesis of relevant studies suggests that higher temperatures in the exercise state are more effective in the induction of irisin.

The paper you recommended was helpful in our understanding of irisin, and we cited it (line 160). Due to the topic and length of the article, we have partially elaborated on irisin-related characteristics in the text (lines 159 to 160 and lines 190 to 193). We hope that our understanding of the relevant literature will answer your questions.

Comments 11: mBDNF (not MBDNF) even after a full stop, line 258 line; otherwise, change in Mature BDNF.

Response 11: Thank you for the reminder.  We changed MBDNF to mBDNF on line 272.

Comments 12:  I would suggest addressing ROS level in relationship to immune system (i.e., ROS accumulation could be harmful for immune system through Nrf2, Wang et al., 2010; Hieltqvist et al., 2007).

Response 12: Thank you very much for your advice. We have read the literature you provided and reviewed the relevant literature. After learning the relationship between ROS and Nrf2, we decided to add relevant statements in lines 400-403 “ROS can inhibit the nuclear translocation of nuclear factorerythroid 2 p45-related factor 2 (Nrf2), destroy the redox homeostasis in macrophages, and lead to the proinflamma-tory response of M1 macrophages” in 4.1.

Comments 13: The Authors assess that this is a systematic review (even if it seems quite narrative) thus Methods should be included.

Response 13: Thank you very much for your advice. There is a deviation in our understanding of systematic review. Now we have changed systematic review to narrative review. Your judgement on the type of our articles is very accurate.

Comments 14: There are several misspelling and grammar should be revised through the text (i.e., the last sentence of figure 2 legend misses the subject).

Response 14: Thank you for the reminder.  Figure 2 has added the subject word "Myokines". In addition, we re-checked the full text and revised several misspelling and grammar, such as the words in lines 377, 379, 387 and 406.

Comments 15: Acronyms should be defined the first time in the text and indicated in parentheses after extended definition, instead there are many acronyms without definitions (i.e., HICT, as high intensity circuit training, 240 line; H0-1, line 389 line; AMPK, line 403, and so on).

Response 15: Thank you very much for your advice. We have added high intensity circuit training (HICT) on line 254, nuclear factorerythroid 2 p45-related factor 2 (Nrf2) and heme oxygenase (HO)-1 were added in lines 401 and 404. AMP-activated protein kinase (AMPK) was added on line 418. Your careful observation makes our article more rigorous.

Reviewer 2 Report

Comments and Suggestions for Authors

Lu and colleagues analyze myokine secretion after exercise and relate it to the effect these myokines have on the immune system to address the beneficial effects of exercise on immune system. The topic is relevant, pertinent and arises very much interest not only in the scientific community but also in health promotion and divulgation. The article is fairly well written, though it requires major adjustments. At some points, authors seem to miss general English language rules. As a reader, I get the impression that not enough care was destined to the writing of this MS, as concepts are not well defined and are very much repeated, loosing conciseness.  

I will provide major commentaries in this section and will highlight minor issues directly in the MS.

Major concerns:

Exercise intensity is alluded throughout the MS. However, it is not clear what is high or moderate intensity. As these vary with each person´s possibilities, I believe a small chart classifying exercise intensity in some way could be clarifying. This should include age and health condition, since this is the aim of the review.

A second issue is the way of presentation. This is a personal opinion and it is up to the authors to take it into consideration, but as a reader, I would prefer to find each myokine discussed in all its aspects: muscle secretion and action and influence over immune system, instead of in two disconnected parts.

Please keep in mind that this article will not be restricted to the specialized reader, thus an extra effort to make all issues clear in required.

Specific comments:

Abstract:

The last sentence: “The aim of this review article is to review the effects of myokines on intrinsic and adaptive immunity and the important role that exercise plays in them, to provide a theoretical basis for the promotion of health by exercise, to provide a potential mechanism for the association between the expression of myokines and immunity as well as the involvement of exercise in organismal immunity, and to offer the possibility of finding suitable exercise training protocols for diseases of the immune system”, is too long, and the last part (indicated in red) is not sufficiently addressed in the MS, i.e., it cannot be claimed as an objective. Therefore, separate it from the rest of the text and offer it as a potential application of the MS, not as an aim.

Text:

Page 1:

“The effects of exercise on the body's immune function are two-fold[2].” Two-fold?

“Some of these exercise-induced myokines have been shown to exert benign effects on the organism, including against chronic and inflammatory diseases such as diabetes and tumours[4], including irisin, IL-6, IL-10, IL-15, brain-derived neurotrophic factor (BDNF), fibroblast growth factors 2 and 21 (FGF2/21), leukaemia inhibitory factor (LIF) and insulin-like growth factor 1 (IGF-1), among others[5-7].” This is too long a sentence with two different clauses (“including”), which should be separated and adequately addressed. Please rewrite.

Page 3:

“NK cells function primarily through the production of various effectors and cytotoxicity, …” various effectors and cytotoxicity are not under the same category and should be adequately addressed.

Page 12:

“The FNDC5 gene is encoded by an atypical initiation codon encoded by an atypical start codon, and its relationship to irisin has been extensively demonstrated.” This sentence needs to be fully rephrased. First: a gene is not encoded by its initiation codon, but by all the codons required to make the gene, usually more than 100, then ENCODED is WRONG. Second, a start codon and an initiation codon are the same thing, thus, you are saying TWICE sth wrong. Third, last part (relation to irisin) is redundant as it has been said in the sentence before, thus is not needed. Actually, the whole sentence can be removed, as it does not make a contribution to the rest of the text.

“In conclusion, current studies generally agree that the appropriate type of exercise (e.g., resistance exercise) and ambient temperature are favourable for inducing the expression of irisin and exerting its potential positive effects.” In this paragraph, the “appropriate” exercise is alluded twice. However, it is not clear for the reader what is appropriate (for whom? Under which circumstances?). Both need to be adequately addressed.

“Members of the interleukin family are small molecular peptides or glycoproteins that can be produced by a variety of cells[57].” Again, this is wrong. Can be replaced by: “Members of the interleukin family are low molecular weight proteins or glycoproteins, …”

However, among these cytokines, IL-6 and …” however is not required, please remove.

Same sentence, “… have been found to be released by the muscle and to exert anti-inflammatory effects as actin, which have a beneficial effect on the organism.” Either explain “as actin” or remove.

Page 13:

“(IL-6 can be released) by skeletal muscle and adipose tissue into the blood to exert endocrine effects. It is also known as lipomyokines[59], which plays a role in regulating metabolism and anti-inflammation.” This needs to be rephrased.

“After a round of quadriceps high-resistance exercises for young men, levels of LIF in the muscle increased significantly”. This means the exercise is only for young men (how could this be so?), we do not know who performed it (and got high LIF levels). Please rephrase.

Page 14:

TrkB stands for Tropomyosin receptor kinase B. Better replace.

“… where phosphorylation activates the Ras-MAPK pathway and finally CREB at the serine site of the cAMP response element binding protein (CREB) [82]. TrkB phosphorylation activates the Ras-MAPK pathway and finally activates CREB at the serine site of the cAMP-responsive element-binding protein (CREB) (51).” This is said twice, please unify.

Page 15:

“Fibroblast growth factors (FGFs) comprise a group of cell signalling proteins with mitogenic activity. It binds to three different types of cell surface molecules, the FGF receptor tyrosine kinase[97], the cysteine-rich FGF receptor[98], and heparan sulphate proteoglycan[99], and is able to promote cell differentiation, migration, and survival[100]. FGF2 is abundant in homogenates of muscle tissue and can be secreted by myotubes cultured in vitro[101].

Two issues in this paragraph:

1.       Better if reserving this paragraph for a general FGF introduction (do not include FGF2). Must change to the plural.

2.       FGF binding to different cell surface molecules: literature should be updated, a modern point of view is needed in regard to interaction partners. Receptor tyrosine kinase speaks of a domain in a family of receptors (the tyrosine kinase domain), not of the whole receptor. In a similar way, cysteine-rich does not define a receptor. Usually, the extracellular moieties of membrane receptors are cysteine rich, whereas the intracellular domain holds TK activity, that is to say, you find both features in the same protein. Thirty years have passed since its initial characterization.

Page 16:

“The IGF-1 molecule consists of four subunits, namely A, B, C, and D. Among these 347 subunits, the structure of the AB subunit bears resemblance to that of insulin[113].” This is wrong. You cannot speak of subunits, IGF1 is a very small protein (only 70 aa), only one chain (subunit). Due to its high homology with insulin, segments of the polypeptide have been assigned with letters corresponding to the insulin counterparts. It is IGF1R that has four subunits similar to Insulin receptor.

“Matheny et al. observed that 16 weeks of resistance training increased liver IGF 1 defect mice quadriceps, calf and feet muscles IGF 1 level.” This is not readily understandable.

Page 18:

It is not clear what exercise skeletal muscle cells are, or what are you referring to. Moreover, the whole sentence is too long and should be cut in two.

Page 19:

“Han et al. found that increased expression of IL-6 induced the expression of IL-6 receptor, p-JAK2 and p-STAT3, …” IL6 induces protein expression or protein phosphorylation (activation in this case). You cannot speak of expression of phosphorylated proteins (indicated by the small p). Please change.

Page 20:

Before the clinical experiment. Nir Yoyev waiting on experimental encephalomyelitis (EAE) in the study of disease found that CD4+ T cells in the IL-10 signal can contribute to the survival of CD4+ T cells.” Please, rewrite, as this is not clear (not understandable).

Page 22:

“A study using recombinant human fibroblast growth factor 21 (rhFGF21) with middle cerebral artery occlusion (MCAO) in mice.” is not a sentence.

Page 23:

“In addition, beyond the effects of exercise, the secretion of actinomycetin may also be related to factors such as diet and aging[182,183].” Actinomycetin???

Color code for highlights in the MS:

Pink and green are discussed in this revision.

Blue (light blue) indicates repetition that should be avoided.

Yellow indicates minor issues discussed in situ (comments in the MS), or typing and formatting issues not commented.

Comments on the Quality of English Language

My comments are included in the preceding section. 

Author Response

1.Summary

Thank you very much for your comments, which has brought great inspiration to us. Based on your suggestions, we have tried our best to make adjustments and believe it will bring great improvements to our articles. The adjustments are as follows.

2.Point-by-point response to Comments and Suggestions for Authors

Comments 1: Exercise intensity is alluded throughout the MS. However, it is not clear what is high or moderate intensity. As these vary with each person´s possibilities, I believe a small chart classifying exercise intensity in some way could be clarifying. This should include age and health condition, since this is the aim of the review.

Response 1: Thank you for pointing this out.  Based on your suggestion, We found the criteria proposed by the American College of Sports Medicine (ACSM) for classifying exercise intensity and drew a small table. ACSM proposed to use HR or VO2max as the criterion for classifying exercise intensity in people of different ages. The references and explanations are given in lines 673-676 in Discussion and conclusion. We include this table as supplementary material. Hopefully, this table will explain the division of exercise intensity more intuitively.

Comments 2: A second issue is the way of presentation. This is a personal opinion and it is up to the authors to take it into consideration, but as a reader, I would prefer to find each myokine discussed in all its aspects: muscle secretion and action and influence over immune system, instead of in two disconnected parts.

Response 2: Thank you very much for your advice. Putting muscle secretion and action and influence over immune system in one section may make the content more coherent. We agree with you. However, all authors discussed the issue of article structure together. We believe that if the content of the article is combined, the various parts about myokines will become lengthy, which may produce a bad reading experience for the reader. Therefore, we wanted to retain the current article structure.

Comments 3: The last sentence: “The aim of this review article is to review the effects of myokines on intrinsic and adaptive immunity and the important role that exercise plays in them, to provide a theoretical basis for the promotion of health by exercise, to provide a potential mechanism for the association between the expression of myokines and immunity as well as the involvement of exercise in organismal immunity, and to offer the possibility of finding suitable exercise training protocols for diseases of the immune system”, is too long, and the last part (indicated in red) is not sufficiently addressed in the MS, i.e., it cannot be claimed as an objective. Therefore, separate it from the rest of the text and offer it as a potential application of the MS, not as an aim.

Response 3: This recommendation is valuable. We have now modified lines 16-21. I changed one sentence to three sentences, And changed the last sentence to "It also provides the possibility to find a suitable exercise training program for immune system diseases."

Comments 4: “The effects of exercise on the body's immune function are two-fold[2].” Two-fold?

Response 4: Thank you very much for your advice. Because English is not our mother tongue, there are some misunderstandings in the translation process. We have changed "two-fold" to "two sides" on line 29.

Comments 5: “Some of these exercise-induced myokines have been shown to exert benign effects on the organism, including against chronic and inflammatory diseases such as diabetes and tumours[4], including irisin, IL-6, IL-10, IL-15, brain-derived neurotrophic factor (BDNF), fibroblast growth factors 2 and 21 (FGF2/21), leukaemia inhibitory factor (LIF) and insulin-like growth factor 1 (IGF-1), among others[5-7].” This is too long a sentence with two different clauses (“including”), which should be separated and adequately addressed. Please rewrite.

Response 5: Thank you very much for your advice. Too long and repetitive sentences can affect the reading experience, so we have adjusted the sentence to "Irisin, interleukin-6/10/15 (IL-6, IL-10, IL-15), brain-derived neurotrophic factor (BDNF)," fibroblast growth factors 2 and 21 (FGF2/21),  leukaemia inhibitory factor (LIF) and in-sulin-like growth factor 1 (IGF-1) and other myokines have been proved to be able to resist chronic and inflammatory diseases, such as diabetes and tumor ", in lines 37-40.

Comments 6: “NK cells function primarily through the production of various effectors and cytotoxicity, …” various effectors and cytotoxicity are not under the same category and should be adequately addressed.

Response 6: Thank you very much for your valuable advice. We reread the citations, and modified to "NK cells mediate cytotoxicity to exert immune effects and regulate other leukocyte subsets of the innate and adaptive immune system through the release of antitumor factors and chemokines ", in lines 97 to 99 in 2. Potential immune cells impacted by exercise.

Comments 7: “The FNDC5 gene is encoded by an atypical initiation codon encoded by an atypical start codon, and its relationship to irisin has been extensively demonstrated.” This sentence needs to be fully rephrased. First: a gene is not encoded by its initiation codon, but by all the codons required to make the gene, usually more than 100, then ENCODED is WRONG. Second, a start codon and an initiation codon are the same thing, thus, you are saying TWICE sth wrong. Third, last part (relation to irisin) is redundant as it has been said in the sentence before, thus is not needed. Actually, the whole sentence can be removed, as it does not make a contribution to the rest of the text.

Response 7: Thank you very much for your careful reminder. Our formulation here is clearly not clear enough. We have resorted the paragraphs from lines 150 to 161 as you suggested, and we have decided to delete this sentence.

Comments 8: “In conclusion, current studies generally agree that the appropriate type of exercise (e.g., resistance exercise) and ambient temperature are favourable for inducing the expression of irisin and exerting its potential positive effects.” In this paragraph, the “appropriate” exercise is alluded twice. However, it is not clear for the reader what is appropriate (for whom? Under which circumstances?). Both need to be adequately addressed.

Response 8: Your advice is very valuable. We recombusted this paragraph and rewrote the sentences in lines 193 to 196. We changed it to "In conclusion, the potential positive effects of irisin can be better played by selecting appropriate exercise types and ambient  temperature according to different genders and diseases." And we draw Supplementary Table 1, hoping to have a better explanation of the definition of exercise intensity.

Comments 9: “Members of the interleukin family are small molecular peptides or glycoproteins that can be produced by a variety of cells[57].” Again, this is wrong. Can be replaced by: “Members of the interleukin family are low molecular weight proteins or glycoproteins, …”

Response 9: This recommendation is valuable. We have changed line 198 to "Members of the interleukin family are low molecular weight proteins or glyco-proteins". Your valuable revision comments make our article more rigorous.

Comments 10: “However, among these cytokines, IL-6 and …” however is not required, please remove.

Response 10: Thank you very much for your advice. We've removed line 199 “However”.

Comments 11: Same sentence, “… have been found to be released by the muscle and to exert anti-inflammatory effects as actin, which have a beneficial effect on the organism.” Either explain “as actin” or remove.

Response 11: Thank you very much for your careful advice. We've removed "as actin" on line 201.

Comments 12: “(IL-6 can be released) by skeletal muscle and adipose tissue into the blood to exert endocrine effects. It is also known as lipomyokines[59], which plays a role in regulating metabolism and anti-inflammation.” This needs to be rephrased.

Response 12: Thank you very much for your advice. We rearrange the sentences on lines 203 to 205, Changed to "IL-6 can be released from skeletal muscle and adipose tissue into the blood to exert endocrine effects. It is also known as lipomyokine and has metabolic and an-ti-inflammatory effects.”

Comments 13: “After a round of quadriceps high-resistance exercises for young men, levels of LIF in the muscle increased significantly”. This means the exercise is only for young men (how could this be so?), we do not know who performed it (and got high LIF levels). Please rephrase.

Response 13: Thank you very much for your suggestion. We reread the cited literature. In this study, Broholm et al. recruited young men as subjects and examined their muscle and blood indices, which contained changes in LIF. To avoid controversy, we have rewritten lines 227-228. The rewrite reads as follows: “Broholm et al. found that subjects had a significant increase in LIF levels in their mus-cles after a round of high-resistance quadriceps training“ in lines 227-228.

Comments 14: TrkB stands for Tropomyosin receptor kinase B. Better replace.

Response 14: Thank you very much for your careful advice. We changed to "Tropomyosin receptor kinase B" in line 272.

Comments 15: “… where phosphorylation activates the Ras-MAPK pathway and finally CREB at the serine site of the cAMP response element binding protein (CREB) [82]. TrkB phosphorylation activates the Ras-MAPK pathway and finally activates CREB at the serine site of the cAMP-responsive element-binding protein (CREB) (51).” This is said twice, please unify.

Response 15: Thank you for reminding me. We have removed the duplicate part, which is detailed in lines 273-276 in 3.3.

Comments 16: “Fibroblast growth factors (FGFs) comprise a group of cell signalling proteins with mitogenic activity. It binds to three different types of cell surface molecules, the FGF receptor tyrosine kinase[97], the cysteine-rich FGF receptor[98], and heparan sulphate proteoglycan[99], and is able to promote cell differentiation, migration, and survival[100]. FGF2 is abundant in homogenates of muscle tissue and can be secreted by myotubes cultured in vitro[101].

Two issues in this paragraph:

  1. Better if reserving this paragraph for a general FGF introduction (do not include FGF2). Must change to the plural.
  2. FGF binding to different cell surface molecules: literature should be updated, a modern point of view is needed in regard to interaction partners. Receptor tyrosine kinase speaks of a domain in a family of receptors (the tyrosine kinase domain), not of the whole receptor. In a similar way, cysteine-rich does not define a receptor. Usually, the extracellular moieties of membrane receptors are cysteine rich, whereas the intracellular domain holds TK activity, that is to say, you find both features in the same protein. Thirty years have passed since its initial characterization.

Response 16: Thank you very much for your advice. First of all, We put the original line 325 "FGF2 is abundant in homogenates of muscle tissue and can be secreted by myo-tubes cultured in vitro" in line 329-330 of the next paragraph. Secondly, through your reminder, we re-reviewed the literature and made changes to the presentation of FGF receptors. We have updated our literature and views. “Fibroblast growth factors (FGFs) are a structurally related family of 22 molecules. FGFs bind to four high-affinity,  ligand-dependent FGF receptor tyrosine kinase mole-cules (FGFR1-4) " in lines 324-328. Your valuable suggestions make our article more rigorous.

Comments 17: “The IGF-1 molecule consists of four subunits, namely A, B, C, and D. Among these 347 subunits, the structure of the AB subunit bears resemblance to that of insulin[113].” This is wrong. You cannot speak of subunits, IGF1 is a very small protein (only 70 aa), only one chain (subunit). Due to its high homology with insulin, segments of the polypeptide have been assigned with letters corresponding to the insulin counterparts. It is IGF1R that has four subunits similar to Insulin receptor.

Response 17: Thank you for reminding me. We totally agree with you. By reviewing the literature, We changed the wording of lines 359-362 to "Insulin-like growth factors (IGFs), including IGF-I and IGF-II," are evolutionarily conserved peptides related to the insulin structure. Mature IGF-I and IGF-II consist of A, B, C,  and D domains. Homology of the A and B domains of insulin-like growth factor to insulin." Your valuable suggestions make our article more meaningful.

Comments 18: “Matheny et al. observed that 16 weeks of resistance training increased liver IGF 1 defect mice quadriceps, calf and feet muscles IGF 1 level.” This is not readily understandable.

Response 18: Thank you for your valuable advice. This is a writing error on our part. On lines 367-369, We have changed it to "Matheny et al. observed that 16 weeks of resistance training increased IGF-1 levels in quadriceps, calf, and foot muscles of liver IGF-1-deficient (LID) mouse ".

Comments 19: “Han et al. found that increased expression of IL-6 induced the expression of IL-6 receptor, p-JAK2 and p-STAT3, …” IL6 induces protein expression or protein phosphorylation (activation in this case). You cannot speak of expression of phosphorylated proteins (indicated by the small p). Please change.

Response 19: This recommendation is valuable.  We re-read this reference. This literature is a study on the expression of phosphorylated proteins. After discussion, we thought that deleting this part would have little effect on the main idea of the paper, so we decided to delete this part.

Comments 20: “Before the clinical experiment. Nir Yoyev waiting on experimental encephalomyelitis (EAE) in the study of disease found that CD4+ T cells in the IL-10 signal can contribute to the survival of CD4+ T cells.” Please, rewrite, as this is not clear (not understandable).

Response 20: Thank you very much for your advice. We couldn't agree more. We have changed the relevant expression in lines 540 to 543 to "In preclinical experiments, Nir Yoyev et al., in their study of experimental encephalo-myelitis (EAE) disease, showed that IL-10 signaling in CD4+T cells can promote CD4+T cell survival,  which enhances autoimmunity in the central nervous system (CNS)."

Comments 21: “A study using recombinant human fibroblast growth factor 21 (rhFGF21) with middle cerebral artery occlusion (MCAO) in mice.” is not a sentence.

Response 21: Thank you very much for reminding me. We rewrote the sentence on lines 632-637, "In one study, recombinant human fibroblast growth factor 21 (rhFGF21) was used to treat middle cerebral artery occlusion (MCAO) mice. Compared with the control group,  the number of CD68+ and CD86+ macrophages in the rhFGF21 treatment group was significantly reduced,  and rhFGF21 inhibited the transformation of macrophages to M1 phenotype and played an anti-inflammatory role. It can  promote the functional recov-ery of stroke rats ".

Comments 22: “In addition, beyond the effects of exercise, the secretion of actinomycetin may also be related to factors such as diet and aging[182,183].” Actinomycetin???

Response 22: Thank you for your careful advice. There was a misunderstanding due to a writing error. We have changed line 692 to "myokines".
